# The Analysis of Genetic Polymorphism on Mitochondrial Hypervariable Region III in Thai Population

**DOI:** 10.3390/genes14030682

**Published:** 2023-03-09

**Authors:** Jirat Vanichanukulyakit, Supakit Khacha-ananda, Tawachai Monum, Phatcharin Mahawong, Kittikhun Moophayak, Watsana Penkhrue, Taddaow Khumpook, Sorawat Thongsahuan

**Affiliations:** 1Department of Forensic Medicine, Faculty of Medicine, Chiang Mai University, Chiang Mai 50200, Thailand; jirat.vanich@cmu.ac.th (J.V.); tawachai.m@cmu.ac.th (T.M.); phatcharin.m@cmu.ac.th (P.M.); 2Research Center in Bioresources for Agriculture, Industry and Medicine, Chiang Mai University, 239, Huay Kaew Road, Muang, Chiang Mai 50200, Thailand; 3Nakhonsawan Campus, Mahidol University, Nakhonsawan 60130, Thailand; khun_khithop@hotmail.com; 4School of Preclinic, Institute of Science, Suranaree University of Technology, Nakhon Ratchasima 30000, Thailand; watsana.p@sut.ac.th; 5Faculty of Science at Sriracha, Kasetsart University, Sriracha Campus, Chonburi 20230, Thailand; taddaow.k@ku.th; 6Faculty of Veterinary Science, Prince of Songkla University, Songkhla 90110, Thailand; sorawat.t@psu.ac.th

**Keywords:** mitochondrial DNA, control region, hypervariable region, Thai population

## Abstract

Mitochondrial DNA (mtDNA) analysis is a genetic marker for human identification, especially matrilineal inheritance. Hypervariable regions (HVR) I and II of mtDNA have been currently performed for human identification worldwide. Further examination of HVRIII has been conducted with the aim of enhancing the power of discrimination. The aim of this research is to provide informative data on the polymorphisms of HVRIII in the Thai population in order to establish a national database for human identification. Thai people who were unrelated through the maternal lineage were recruited for blood collections. The mtDNA was extracted by Chelex extraction, amplified by polymerase chain reaction, and analyzed using Sequencing Analysis Software. The most common mutation in HVRIII was base substitution, followed by deletion and insertion. We discovered 40 unique haplotypes, with haplotype 489C being the most frequent. The haplotype diversity, power of discrimination, and random match probability were 0.8014, 0.7987, and 0.2013, respectively. Five-CA repeats were the most frequently observed in nucleotide positions 514–523. Our database can be employed as supplementary markers in addition to nuclear deoxyribonucleic acid (DNA) markers in forensic investigations. Moreover, the data could potentially enhance genetic identification and anthropological genetics research in Thailand.

## 1. Introduction

Human identification plays an important role in the criminal process. The deoxyribonucleic acid (DNA) fingerprint of the suspected person was compared with identified DNA derived from biological remains at the scene or with the DNA of other people for the objective of identification or exclusion [1]. Intracellular DNA originates from the nucleus (nuclear DNA) and mitochondria (mitochondrial DNA, or mtDNA) [2]. The mitochondrial genome is divided into a lengthy coding area and a small region that has a highly polymorphic non-coding section known as the control region or D-loop [3]. The control region can be divided into three sections: the hypervariable region I (HVRI), located between nucleotide positions of 16,024 and 16,365; the hypervariable region II (HVRII), ranging from the nucleotide positions of 73 and 340; and the hypervariable region III (HVRIII), extending between the nucleotide positions of 438 and 574 [4]. The difference between nuclear and mitochondrial DNA is the inheritance pattern. Nuclear DNA is inherited equally from both parents, whereas mitochondrial DNA is uniquely inherited from maternal relatives. MtDNA does not undergo recombination and is only passed exclusively maternally to the offspring [5]. Additionally, the copy number per cell of mitochondrial DNA is higher than that of nuclear DNA [6]. The mitochondrial DNA is also less affected by specimen degradation than the nuclear DNA [7,8]. Hence, mtDNA sequence has been shown to be an effective and trustworthy tool for forensic biometric identification in cases in which inadequate amounts and quality of nuclear DNA exist [9,10,11]. Moreover, the great advantage of mtDNA typing in human identification is the high mutation rate compared to nuclear DNA, resulting in an increase in the power of discrimination between non-blood relationships and exerting high individual variation. High mutation in mtDNA derived from two events. First, mitochondria are the energy production of the cells. During the oxidative phosphorylation process, numerous oxygen radicals were produced and directly caused oxidative damage to mtDNA, causing mutations to accumulate over time. Second, the proofreading activity of DNA polymerase during the replication process in mtDNA was inefficient, which cannot be repaired for some mutated bases [12,13]. 

According to the Scientific Working Group on DNA Analysis Methods (SWGDAM), the analysis of mtDNA sequence is based on the typing of both HVRI and HVRII polymorphisms. If the mtDNA polymorphism between two samples is identical, the samples cannot be excluded. However, mtDNA typing could not be concluded when the difference of one base mutation on both HVRI and HVRII was found [14]. Moreover, a highly common polymorphism in these two regions (HVRI and HVRII) affected presenting a similar sequence in more than one individual [15]. To confirm relationships between individuals presenting only one difference in the sequences of HVRI and HVRII as well as increase the power of discrimination, supplementary analysis of HVRIII may be helpful. To present the probability of maternity between individuals, the likelihood ratio was calculated by examining the frequency of observed haplotypes in the hypervariable region within their specific population [15,16]. However, the geographic location and ethnicity of the population have been reported to influence the distribution of mtDNA polymorphism [17,18,19,20]. Comas et al., 1996 summarized that a geographic variation had an impact on a different distribution of mtDNA control regions among European populations [21]. He et al., 2014 found a variation in mtDNA polymorphism among each ethnic group in the Chinese population; moreover, these variations were considerably different from those in other Asian populations [22]. According to Thai historical records, the current Thai population is a genetic admixture due to an intermix between several foreign migrations [23]. Therefore, the polymorphism database of mtDNA for person identification can obviously be used in specific populations or individual countries for precise typing in forensic casework. Herein, the goal of this research is to provide informative data on the polymorphisms of HVRIII regions on mtDNA in the Thai population in order to establish a national database for human identification by mtDNA typing.

## 2. Materials and Methods

### 2.1. Sample Collection

Three hundred and two blood samples were collected from unrelated individuals through the maternal lineage who resided in Thailand. Thai nationality was identified by a Thai identification card. All participants were endorsed with informed consent documents. The maternal-related history was interviewed prior to blood collection to confirm that their maternal ancestors were of Thai descent. This research has been approved by the Ethics Committee of the Faculty of Medicine, Chiang Mai University, Chiang Mai, Thailand (code. FOR-2564-08419). Our samples were collected from participants who lived in all regions of Thailand, consisting of the Northern, Northeastern, Western, Central, Eastern, and Southern regions. The number of participants who endorsed our research is presented in Table 1.

### 2.2. DNA Extraction

DNA from dried blood in cotton bud tips was extracted by the Chelex extraction method [24]. The dried blood on cotton bud tips was placed in a microcentrifuge tube, and sterile water was added. After incubation for 1 h, the solution was centrifuged at 13,000 rpm for 5 min to collect the cells. The supernatant was removed, and the Chelex chelating resin was then added into the tube. The solution was incubated at 56 °C for at least 1 h, followed by 100 °C for 8 min. The extracted DNA was kept at −20 °C prior to the analysis of mtDNA typing.

### 2.3. Hypervariable Region III (HVRIII) Amplification

Two primer sets were used to amplify HVRIII (F162 5′ CGCACCTACGTTCAATATTACA 3′ and R638 5′ GGTGATGTGAGCCCGTCTAA 3′) [21]. Polymerase chain reaction (PCR) to amplify specific regions on the DNA strand was performed in a microcentrifuge with a genomic DNA template, mixed primer, and JumpStart™ Taq DNA Polymerase according to the manufacturer’s instructions (Sigma-Aldrich, St. Louis, MO, USA). The amplification was carried out by an Analytik-Jena FlexCycler (Analytik-Jena, Jena, Germany) with cycling conditions of 94 °C for 90 s, 39 cycles of 53 °C for 10 s, and 72 °C for 72 s. After amplification, DNA amplicons were purified by 100% isopropanol precipitation and dissolved with 10 mM Tris-HCl prior to being used as a substrate in nucleotide sequencing.

### 2.4. Nucleotide Sequencing

To sequence the nucleotide on HVRIII by Sanger technique, the purified amplicon in Tris-hydrochloric acid solution was mixed with 3.2 mM primer, dilution buffer, and BigDye terminator version 3.1 according to manufacturer instructions (ThermoFisher Scientific Inc., MA, USA). Thermal cycling was used to amplify the signal, starting at 96 °C for 70 s, going through 24 cycles of 50 °C for 5 s, 60 °C for 240 s, and ending with a hold at 56 °C. Before being analyzed on an ABI 3500 Genetic Analyzer (ThermoFisher Scientific Inc., Waltham, MA, USA), the amplified product reaction was purified by absolute ethanol (99.8%) precipitation, and the purified nucleotide sequences were then mixed with LIZ-HiDi solution.

### 2.5. Data Analysis

The nucleotide sequences were analyzed using Sequencing Analysis Software v6.0 [25] and then aligned to compare with the Revised Cambridge Reference Sequences (rCRS) [26] using SeqScape™ Software version 3.0 [25,27]. The number of mutations on HVRIII was expressed as the observed number and percentage of observations. To calculate forensic parameters, the random match probability (RMP), power of discrimination (PD), and haplotype diversity (HD) [28] were calculated as follows:$$Random\;match\;probability\;\left(RMP\right)=\Sigma \left(X^{2}\right)\;Power\;of\;discrimination\;\left(PD\right)=1-\Sigma \left(X^{2}\right)\;Haplotype\;diversity\;\left(HD\right)=\frac{\left(1-\Sigma X^{2}\right)n}{n-1}$$
where “X” refers to the observed haplotype’s frequency and “n” refers to the number of populations.

## 3. Results and Discussion

After nucleotide sequencing, the mutations in the class of substitutions, insertions, and deletions on hypervariable region III (HVRIII) were recorded. We found 28 variable sites in HVRIII. The most common mutation on HVRIII was base substitution, followed by deletion and insertion. Regarding base substitutions, it was discovered that nucleotide transitions predominated over nucleotide transversion. The transition from thymine to cytosine (T→C) was shown to be the most frequent nucleotide substitution, followed by the transitions from cytosine to thymine (C→T), guanine to adenine (G→A), and adenine to guanine (A→G), respectively. The highest substitutions were found at the nucleotide position of 489 thymine to cytosine (T→C), at 79.70%. An adenine’s nuclear transversion to cytosine was shown at the nucleotide position of 574. A cytosine’s nuclear transversions to guanine were additionally shown at the nucleotide positions of 447 and 530. In contrast, no transversion between cytosine and adenine (C→A), thymine and guanine (T→G), thymine and adenine (T→A), and guanine and cytosine (G→C) was found in this study. The result is shown in Table 2. A nucleotide mutation at a nucleotide position of 489 (T→C) is believed to be unique to southeast Asian groups rather than other populations, according to previous research [20].

The nucleotide insertions were observed at three sites, including at the nucleotide positions of 514, 523, and 573 (with +AC at the nucleotide position of 514, +CA at the nucleotide position of 523, +CACA the nucleotide position of 523, +C at the nucleotide position of 573, and +CC at the nucleotide position of 573). The most common insertion site was +CA at the nucleotide position of 523, which was found in 7 of 302 participants in this study. Additionally, nucleotide deletion sites were also observed in seven sites, with −A at the nucleotide positions of 521 and 523, −G at the nucleotide position of 513, and −C at the nucleotide positions of 459, 514, 520, and 522. The nucleotide deletion (CA and −CACA (−2CA)) at a nucleotide position between 520 and 523 was interpreted in accordance with a previous study [29]. The two most prevalent nucleotide deletion sites in this study’s 302 participants were −CA at the nucleotide positions between 522 and 523 (found in 127 of 302) and −CACA (−2CA) at the nucleotide positions between 520 and 523 (found in 3 of 302). The result is shown in Table 2.

The comparison of mtDNA polymorphism between different regions of Thailand is shown in Table 3. We found the number of total variation sites among the participants who lived in the Northern, Northeast, West, Central, Eastern, and Southern regions of Thailand to be 13, 11, 7, 9, 8, and 12, respectively. Despite being the top two most frequent nucleotide variations in this population, transition T→C at the nucleotide position of 489 and del522-del523 were also discovered to be the top two most frequent nucleotide variations in each of the population’s regions. In the North, West, Central, and Southern regions, transition T→C at the nucleotide position of 489 was found to be the most common nucleotide variation, and del522-del523 was found to be the second most common. In the Northeastern and Eastern regions, del522-del523 was the most observed nucleotide variation, and transition T→C at the nucleotide position of 489 was followed. However, in each region, other nucleotide variations were found differently. Moreover, we found the nucleotide transversion (A→C and C→G) in populations from the Northeast and Southern regions only. All studied types of base insertion (+AC, +C, +CC, +CA, and +CACA) were not observed in the population from the Central region of Thailand in our study. Moreover, the population from Northeastern Thailand demonstrated various types of base insertion compared to other regions (+AC at the nucleotide position of 514, +CC at the nucleotide position of 573, +CA at the nucleotide position of 523, and +CACA at the nucleotide position of 523). The deletion of guanine at the nucleotide position of 531 was only observed in the Northern population in our study. Many factors have been reported to affect genetic polymorphism, especially microevolution, such as migration, selection, gene drift, and mutation [30]. Furthermore, the origins of the ancestors of each region originated from different areas, which also affected the genetic variations in our population [31]. 

All mutate observations were grouped to represent the mtDNA haplotype. Forty haplotypes were classified, with 14 haplotypes shared by many individuals and 26 haplotypes unique to an individual, as shown in Table 4. According to the results, haplotype 489C (30.46% of Thai participants in this study) was the most often observed, followed by haplotypes del522-del523 (28.80%), HVRIII-rCRS (no nucleotide mutation was found in HVRIII compared to the Revised Cambridge Reference Sequences (rCRS)), (11.26%), 489C-del522-del523 (10.60%), and 456T-489C (2.32%). Additionally, haplotype 489C-ins523CA was found in four individuals; haplotype ins523CA was found in four individuals; haplotype 485C-489C was found in three individuals; haplotype del520-del523 was found in three individuals; haplotype 488G-489C was found in two individuals; haplotype 489C-518T was found in two individuals; haplotype 489C-573.1C was found in two individuals; haplotype 499A was found in two individuals; and haplotype 513A was found in two individuals. The study identified the following unique haplotypes: 444G, 447G-489C, 454C-489C-del522-del523, 456T, 456T-del522-del523, del459-489C, 461T-489C, 482C, 485C-del522-del523, 488G-489C-del522-del523, 489C-501T-del522-del523, 489C-508G, 489C-513A, 489C-513del-514del, 489C-ins523CACA, 489C-547G, 489C-ins573CC, 489C-574C, 498T-499A, 513A-ins514AC, del513-del514, 518T, del522-del523-530G, del522-del523-533G, and 539C.

After analysis and computation, the haplotype diversity (HD) was 0.8014, the power of discrimination (PD) was 0.7987, and the random match probability (RMP) was 0.2013. In the case of haplotype diversity of 0.8014 and discrimination power of 0.7987, if two individuals were sampled. Their haplotypes’ likelihood of being different was 80.14%, and a selected individual will have a chance of 79.87% of being discriminated against by other individuals in the population [28,32]. The random match probability (RMP) represents the likelihood of two unrelated individuals having the same mtDNA profile by chance [33]. Our calculated RMP was consistent with the findings reported by Thongngam et al., 2016, who conducted research on a sample of 100 unrelated Thai individuals [20]. Previous research in the European population showed the RMP ranged between 0.0116–0.044 [34]. Additionally, our finding provides higher haplotype diversity and a higher power of discrimination than in the previous studies in Japanese and Thai populations. Nagai et al., 2004 showed the haplotype diversity and power of discrimination at 0.782 and 0.777% in 150 unrelated Japanese individuals [18]. Regarding the Thai population, a previous study discovered that the haplotype diversity was 0.643 and the power of discrimination was 0.630 for 100 unrelated Thai individuals [20]. As per the description of these forensic parameters, the increased value of these parameters may be helpful for enhancing genetic identification capabilities. The genetic parameter computed from HVRIII alone does not reach the same level as the results from the study of HVRI, HVRII, or HVRI + HVRII due to shorter nucleotide sequences and a lower number of polymorphic sites [19,35]. The earlier research, however, demonstrated that including HVRIII in addition to HVRI and HVRII in the computation improved results for genetic identification and could distinguish nucleotide sequences that were identical in the HVRI and HVRII regions [15,16].

Additionally, the region between the nucleotide positions of 514 and 523 had five pairs of CA/AC residues, which were only found on HVRIII. The insertion or deletion (indels) of CA or AC bases can be found in this position [29,36]. It is noteworthy that CA/AC residues represented length heteroplasmy or variables similar to autosomal short tandem repeat (STR) [37]. Table 5 displays the “CA repeats” that the Thai participants observed. Three-CA repeats, four-CA repeats, five-CA repeats (no insertion and deletion of rCRS), six-CA repeats, and seven-CA repeats were grouped into five groups in this study and were found in 3, 127, 163, 7, and two participants, respectively. The most frequent repeats discovered in our cohort—as well as the most frequent repeats discovered in previous research from Indian, Malaysian, Sinhalese, Japanese, and Korean populations—were the five-CA repeats (rCRS), and the four-CA repeats were the second-most frequent repeats found in these previous studies [18,20,38,39]. Table 6 shows the observed CA repeats of Thai individuals in each region. The five-CA repeats (no insertion and deletion of rCRS) were most frequently found in the Northern, Western, Central, Eastern, and Southern groups. The four-CA repeats were the second-most frequent CA repeats in these groups. However, in the Northeastern, we found that the most frequent repeats were the four-CA repeats, and the five-CA (no insertion and deletion of rCRS) repeats were the second-ranked. We hypothesized that the heterogeneity of the recruited participants might be the cause of the Northeast’s higher four-CA repeat prevalence than five-CA repeat prevalence. According to two studies in India, four-CA repeats were most frequently found in one study [36], while five-CA repeats were more typically detected in another study [40]. Even in the earlier study in the Thai population, four-CA repeats were the most frequently observed [20]. It might result from inheriting the genetic heritage of ancestors who are the origins of people in the Northeast, though this may need further research to be confirmed. 

In the case of a nucleotide transition at the nucleotide position of 513, the polymorphisms in mitochondrial DNA hypervariable region III at the nucleotide position between 513 and 524 will be prolonged as AC repeats when the nucleotide transition guanine to adenine (G→A) occurs at position 513 [41]. The reference sequence of rCRS has guanine at the nucleotide position of 513 and CA repeats at the nucleotide position between 514 and 523; however, if the 513A mutation occurs, the nucleotide sequences at the nucleotide positions between 513 and 524 will be read as AC repeats. In this study, four individuals with 513A mutations were found, including haplotypes 489C-513A (one participant), 513A (two participants), and 513A-ins514AC (one participant). As six AC repeats, six AC repeats, and seven AC repeats, respectively, the haplotypes can be categorized.

## 4. Conclusions

We examined the nucleotide polymorphism of the mitochondrial DNA hypervariable region III in non-related Thai individuals. It was discovered that nucleotide transitions predominated over nucleotide transversions (a ratio of 194:3). The most frequent substitution was found to be a nucleotide transition from thymine to cytosine (83.25%). The most frequent haplotypes found were 489C (30.46% of Thai participants in this study). The CA repeats were divided into five groups, which were three-CA repeats, four-CA repeats, five-CA repeats (rCRS), six-CA repeats, and seven-CA repeats. Five-CA repeats (rCRS) were found to be the most common CA repeats in our research. The haplotype diversity (HD) was 0.8014, the power of discrimination (PD) was 0.7987, and the random match probability (RMP) was 0.2013. In the case of multiple individuals having the same HVRI and HVRII sequences, HVRIII can provide the supplementary marker required for the forensic identification of a specific person.

## Figures and Tables

**Table 1 genes-14-00682-t001:** The number of participants from each region of Thailand in this research.

Characteristics		Number of Participants					
		Northern	Northeastern	Western	Central	Eastern	Southern
Sex	Male	5	18	17	23	6	16
	Female	39	43	36	39	25	35
Total		44	61	53	62	31	51

**Table 2 genes-14-00682-t002:** Nucleotide sequence polymorphisms in the HVRIII of mitochondrial DNA from 302 unrelated Thai individuals.

Type of Nucleotide Change	Observed Number	Different Nucleotide Positions in HVRIII (Observed Number)
Total number of variant sites	28	
Total number of base substitutions	197	
Transition	194	
A→G	8	444 (1), 488 (3), 503 (1), 508 (1), 533 (1), 547 (1)
G→A	6	499 (2), 513 (4)
T→C	164	454 (1), 482 (1), 485 (4), 489 (157), 539 (1)
C→T	16	456 (9), 461 (1), 498 (2), 501 (1), 518 (3)
Transversion	3	
A→C	1	574 (1)
C→G	2	447 (1), 530 (1)
Total number of base insertions	13	
+AC	1	514 (1)
+C	2	573 (2)
+CC	1	573 (1)
+CA	7	523 (7)
+CACA	2	523 (2)
Total number of base deletions	271	
−A	133	521 (3), 523 (130)
−G	2	513 (2)
−C	136	459 (1), 514 (2), 520 (3), 522 (130)

**Table 3 genes-14-00682-t003:** Nucleotide sequence polymorphisms in the HVRIII of mitochondrial DNA from unrelated Thai individuals from six regions.

Type of Nucleotide Change	Different Nucleotide Position in HVRIII (Observed Number)					
	Northern (*n* = 44)	Northeast (*n* = 61)	Western (*n* = 53)	Central (*n* = 62)	Eastern (*n* = 31)	Southern (*n* = 51)
Total number of base substitutions						
Transition						
A→G	488 (2)	488 (1), 547 (1)		508 (1)		444 (1), 503 (1), 533 (1)
G→A	499 (1)	513 (2)		513 (1)		499 (1), 513 (1)
T→C	485 (1), 489 (23)	489 (25)	489 (34), 454 (1)	489 (35)	485 (2), 489 (14), 539 (1)	482 (1), 485 (1), 489 (26)
C→T	456 (2), 498 (2), 518 (1)	501 (1)	456 (2), 518 (1)	456 (4), 518 (1)	461 (1)	456 (1)
Transversion						
A→C		574 (1)				
C→G		530 (1)				447 (1)
Total number of base insertions						
+AC		514 (1)				
+C	573 (1)		573 (1)			
+CC		573 (1)				
+CA	523 (1)	523 (1)	523 (3)		523 (1)	523 (1)
+CACA		523 (1)				523 (1)
Total number of base deletions						
−A	523 (17)	523 (30)	523 (18)	521 (1), 523 (27)	521 (2), 523 (15)	523 (23)
−G	513 (2)					
−C	459 (1), 514(2), 522 (17)	522 (30)	522 (18)	520 (1), 522 (27)	520 (2), 522 (15)	522 (23)

**Table 4 genes-14-00682-t004:** Nucleotide sequence polymorphisms in the mitochondrial DNA hypervariable region III of 302 non-maternal-related Thai participants. A highlight bar demonstrated the haplotype of HVRIII-rCRS which had no nucleotide mutation compared to the rCRS. * represents no mutation compared to the rCRS.

444	447	454	456	459	461	482	485	488	489	498	499	501	503	508	513	514	514.1	514.2	518	520	521	522	523	523.1	523.2	523.3	523.4	530	533	539	547	573	573.1	573.2	574	n
A	C	T	C	C	C	T	T	A	T	C	G	C	A	A	G	C	-	-	C	C	A	C	A	-	-	-	-	C	A	T	A	C	-	-	A	
*****	*****	*****	*****	*****	*****	*****	*****	*****	**C**	*****	*****	*****	*****	*****	*****	*****	*****	*****	*****	*****	*****	*****	*****	*****	*****	*****	*****	*****	*****	*****	*****	*****	*****	*****	*****	**92**
*****	*****	*****	*****	*****	*****	*****	*****	*****	*****	*****	*****	*****	*****	*****	*****	*****	*****	*****	*****	*****	*****	**del**	**del**	*****	*****	*****	*****	*****	*****	*****	*****	*****	*****	*****	*****	**87**
*****	*****	*****	*****	*****	*****	*****	*****	*****	*****	*****	*****	*****	*****	*****	*****	*****	*****	*****	*****	*****	*****	*****	*****	*****	*****	*****	*****	*****	*****	*****	*****	*****	*****	*****	*****	**34**
*****	*****	*****	*****	*****	*****	*****	*****	*****	**C**	*****	*****	*****	*****	*****	*****	*****	*****	*****	*****	*****	*****	**del**	**del**	*****	*****	*****	*****	*****	*****	*****	*****	*****	*****	*****	*****	**32**
*****	*****	*****	**T**	*****	*****	*****	*****	*****	**C**	*****	*****	*****	*****	*****	*****	*****	*****	*****	*****	*****	*****	*****	*****	*****	*****	*****	*****	*****	*****	*****	*****	*****	*****	*****	*****	**7**
*****	*****	*****	*****	*****	*****	*****	*****	*****	**C**	*****	*****	*****	*****	*****	*****	*****	*****	*****	*****	*****	*****	*****	*****	**C**	**A**	*****	*****	*****	*****	*****	*****	*****	*****	*****	*****	**4**
*****	*****	*****	*****	*****	*****	*****	*****	*****	*****	*****	*****	*****	*****	*****	*****	*****	*****	*****	*****	*****	*****	*****	*****	**C**	**A**	*****	*****	*****	*****	*****	*****	*****	*****	*****	*****	**4**
*****	*****	*****	*****	*****	*****	**C**	*****	*****	**C**	*****	*****	*****	*****	*****	*****	*****	*****	*****	*****	*****	*****	*****	*****	*****	*****	*****	*****	*****	*****	*****	*****	*****	*****	*****	*****	**3**
*****	*****	*****	*****	*****	*****	*****	*****	*****	*****	*****	*****	*****	*****	*****	*****	*****	*****	*****	*****	**del**	**del**	**del**	**del**	*****	*****	*****	*****	*****	*****	*****	*****	*****	*****	*****	*****	**3**
*****	*****	*****	*****	*****	*****	*****	*****	**G**	**C**	*****	*****	*****	*****	*****	*****	*****	*****	*****	*****	*****	*****	*****	*****	*****	*****	*****	*****	*****	*****	*****	*****	*****	*****	*****	*****	**2**
*****	*****	*****	*****	*****	*****	*****	*****	*****	**C**	*****	*****	*****	*****	*****	*****	*****	*****	*****	**T**	*****	*****	*****	*****	*****	*****	*****	*****	*****	*****	*****	*****	*****	*****	*****	*****	**2**
*****	*****	*****	*****	*****	*****	*****	*****	*****	**C**	*****	*****	*****	*****	*****	*****	*****	*****	*****	*****	*****	*****	*****	*****	*****	*****	*****	*****	*****	*****	*****	*****	*****	**C**	*****	*****	**2**
*****	*****	*****	*****	*****	*****	*****	*****	*****	*****	*****	**A**	*****	*****	*****	*****	*****	*****	*****	*****	*****	*****	*****	*****	*****	*****	*****	*****	*****	*****	*****	*****	*****	*****	*****	*****	**2**
*****	*****	*****	*****	*****	*****	*****	*****	*****	*****	*****	*****	*****	*****	*****	**A**	*****	*****	*****	*****	*****	*****	*****	*****	*****	*****	*****	*****	*****	*****	*****	*****	*****	*****	*****	*****	**2**
**G**	*****	*****	*****	*****	*****	*****	*****	*****	*****	*****	*****	*****	*****	*****	*****	*****	*****	*****	*****	*****	*****	*****	*****	*****	*****	*****	*****	*****	*****	*****	*****	*****	*****	*****	*****	**1**
*****	**G**	*****	*****	*****	*****	*****	*****	*****	**C**	*****	*****	*****	*****	*****	*****	*****	*****	*****	*****	*****	*****	*****	*****	*****	*****	*****	*****	*****	*****	*****	*****	*****	*****	*****	*****	**1**
*****	*****	**C**	*****	*****	*****	*****	*****	*****	**C**	*****	*****	*****	*****	*****	*****	*****	*****	*****	*****	*****	*****	**del**	**del**	*****	*****	*****	*****	*****	*****	*****	*****	*****	*****	*****	*****	**1**
*****	*****	*****	**T**	*****	*****	*****	*****	*****	*****	*****	*****	*****	*****	*****	*****	*****	*****	*****	*****	*****	*****	*****	*****	*****	*****	*****	*****	*****	*****	*****	*****	*****	*****	*****	*****	**1**
*****	*****	*****	**T**	*****	*****	*****	*****	*****	*****	*****	*****	*****	*****	*****	*****	*****	*****	*****	*****	*****	*****	**del**	**del**	*****	*****	*****	*****	*****	*****	*****	*****	*****	*****	*****	*****	**1**
*****	*****	*****	*****	**del**	*****	*****	*****	*****	**C**	*****	*****	*****	*****	*****	*****	*****	*****	*****	*****	*****	*****	*****	*****	*****	*****	*****	*****	*****	*****	*****	*****	*****	*****	*****	*****	**1**
*****	*****	*****	*****	*****	**T**	*****	*****	*****	**C**	*****	*****	*****	*****	*****	*****	*****	*****	*****	*****	*****	*****	*****	*****	*****	*****	*****	*****	*****	*****	*****	*****	*****	*****	*****	*****	**1**
*****	*****	*****	*****	*****	*****	**C**	*****	*****	*****	*****	*****	*****	*****	*****	*****	*****	*****	*****	*****	*****	*****	*****	*****	*****	*****	*****	*****	*****	*****	*****	*****	*****	*****	*****	*****	**1**
*****	*****	*****	*****	*****	*****	*****	**C**	*****	*****	*****	*****	*****	*****	*****	*****	*****	*****	*****	*****	*****	*****	**del**	**del**	*****	*****	*****	*****	*****	*****	*****	*****	*****	*****	*****	*****	**1**
*****	*****	*****	*****	*****	*****	*****	*****	**G**	**C**	*****	*****	*****	*****	*****	*****	*****	*****	*****	*****	*****	*****	**del**	**del**	*****	*****	*****	*****	*****	*****	*****	*****	*****	*****	*****	*****	**1**
*****	*****	*****	*****	*****	*****	*****	*****	*****	**C**	*****	*****	**T**	*****	*****	*****	*****	*****	*****	*****	*****	*****	**del**	**del**	*****	*****	*****	*****	*****	*****	*****	*****	*****	*****	*****	*****	**1**
*****	*****	*****	*****	*****	*****	*****	*****	*****	**C**	*****	*****	*****	*****	**G**	*****	*****	*****	*****	*****	*****	*****	*****	*****	*****	*****	*****	*****	*****	*****	*****	*****	*****	*****	*****	*****	**1**
*****	*****	*****	*****	*****	*****	*****	*****	*****	**C**	*****	*****	*****	*****	*****	**A**	*****	*****	*****	*****	*****	*****	*****	*****	*****	*****	*****	*****	*****	*****	*****	*****	*****	*****	*****	*****	**1**
*****	*****	*****	*****	*****	*****	*****	*****	*****	**C**	*****	*****	*****	*****	*****	**del**	**del**	*****	*****	*****	*****	*****	*****	*****	*****	*****	*****	*****	*****	*****	*****	*****	*****	*****	*****	*****	**1**
*****	*****	*****	*****	*****	*****	*****	*****	*****	**C**	*****	*****	*****	*****	*****	*****	*****	*****	*****	*****	*****	*****	*****	*****	**C**	**A**	**C**	**A**	*****	*****	*****	*****	*****	*****	*****	*****	**1**
*****	*****	*****	*****	*****	*****	*****	*****	*****	**C**	*****	*****	*****	*****	*****	*****	*****	*****	*****	*****	*****	*****	*****	*****	*****	*****	*****	*****	*****	*****	*****	**G**	*****	*****	*****	*****	**1**
*****	*****	*****	*****	*****	*****	*****	*****	*****	**C**	*****	*****	*****	*****	*****	*****	*****	*****	*****	*****	*****	*****	*****	*****	*****	*****	*****	*****	*****	*****	*****	*****	*****	**C**	**C**	*****	**1**
*****	*****	*****	*****	*****	*****	*****	*****	*****	**C**	*****	*****	*****	*****	*****	*****	*****	*****	*****	*****	*****	*****	*****	*****	*****	*****	*****	*****	*****	*****	*****	*****	*****	*****	*****	**C**	**1**
*****	*****	*****	*****	*****	*****	*****	*****	*****	*****	**T**	**A**	*****	*****	*****	*****	*****	*****	*****	*****	*****	*****	*****	*****	*****	*****	*****	*****	*****	*****	*****	*****	*****	*****	*****	*****	**1**
*****	*****	*****	*****	*****	*****	*****	*****	*****	*****	*****	*****	*****	**G**	*****	*****	*****	*****	*****	*****	*****	*****	**del**	**del**	*****	*****	*****	*****	*****	*****	*****	*****	*****	*****	*****	*****	**1**
*****	*****	*****	*****	*****	*****	*****	*****	*****	*****	*****	*****	*****	*****	*****	**A**	*****	**A**	**C**	*****	*****	*****	*****	*****	*****	*****	*****	*****	*****	*****	*****	*****	*****	*****	*****	*****	**1**
*****	*****	*****	*****	*****	*****	*****	*****	*****	*****	*****	*****	*****	*****	*****	**del**	**del**	*****	*****	*****	*****	*****	*****	*****	*****	*****	*****	*****	*****	*****	*****	*****	*****	*****	*****	*****	**1**
*****	*****	*****	*****	*****	*****	*****	*****	*****	*****	*****	*****	*****	*****	*****	*****	*****	*****	*****	**T**	*****	*****	*****	*****	*****	*****	*****	*****	*****	*****	*****	*****	*****	*****	*****	*****	**1**
*****	*****	*****	*****	*****	*****	*****	*****	*****	*****	*****	*****	*****	*****	*****	**del**	**del**	*****	*****	*****	*****	*****	*****	*****	*****	*****	*****	*****	**G**	*****	*****	*****	*****	*****	*****	*****	**1**
*****	*****	*****	*****	*****	*****	*****	*****	*****	*****	*****	*****	*****	*****	*****	**del**	**del**	*****	*****	*****	*****	*****	*****	*****	*****	*****	*****	*****	*****	**G**	*****	*****	*****	*****	*****	*****	**1**
*****	*****	*****	*****	*****	*****	*****	*****	*****	*****	*****	*****	*****	*****	*****	*****	*****	*****	*****	*****	*****	*****	*****	*****	*****	*****	*****	*****	*****	*****	**C**	*****	*****	*****	*****	*****	**1**

**Table 5 genes-14-00682-t005:** Insertion and deletion in mitochondrial DNA hypervariable region III at the nucleotide position between 514 and 523 from 302 Thai participants.

Nucleotide Polymorphism in HVRIII at Positions 514–523	Total Number of Nucleotide Changes	% Insertion and Deletion from Samples
CACACA (−CACA)	3	0.99
CACACACA (−CA)	127	42.05
CACACACACA	163	53.97
CACACACACACA (+CA)	7	2.32
CACACACACACACA (+CACA)	2	0.66

**Table 6 genes-14-00682-t006:** Comparisons of observed CA repeats between Thai participants from each region.

Nucleotide Polymorphism in HVRIII at Positions 514–523	Northern	Northeastern	Western	Central	Eastern	Southern
CACACA (−CACA)	0 (0%)	0 (0%)	0 (0%)	1 (1.61%)	2 (6.45%)	0 (0%)
CACACACA (−CA)	17 (38.63%)	30 (49.18%)	18 (33.96%)	26 (41.93%)	13 (41.94%)	23 (45.10%)
CACACACACA	26 (59.09%)	29 (47.54%)	32 (60.38%)	35 (56.45%)	15 (48.39%)	26 (50.98%)
CACACACACACA (+CA)	1 (2.27%)	1 (1.64%)	3 (5.66%)	0 (0%)	1 (3.23%)	1 (1.96%)
CACACACACACACA (+CACA)	0 (0%)	1 (1.64%)	0 (0%)	0 (0%)	0 (0%)	1 (1.96%)

## Data Availability

Not applicable.

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
