# Peer review of "The Analysis of Genetic Polymorphism on Mitochondrial Hypervariable Region III in Thai Population"

_genes, 2023, doi:10.3390/genes14030682_

Round 1
Reviewer 1 Report
In this study, the author studied by collecting blood from the Thai people who were unrelated through the maternal lineage. Aim to provide informative data on the polymorphisms of HVIII in the Thai population to establish a national database for human identification, and the data potentially enhance genetic identification and anthropological genetics research in Thailand. The paper is generally well written and structured. However, in my opinion, the paper needs minor revisions as suggested below.
1. The keywords: “Thai” to “Thai Population”.
2. Line 48: mtDNA should be MtDNA.
3. Line 97: Corresponding references need to be added after the method.
4. Line 104: “hypervariable region III” shoule be “HVIII”.
5. Line 120: What’s the alcohol strength used? Please state the details.
6. Line 121: “Hidi solution” to “LIZ-HiDi solution”
7. Line 124: The relevant reference need to be cited after software, and no company name need to be added.
8. Line 138: add “of ” after Southern.
9. Table 1 need move to the part about sample collection.
Author Response
We would like to thank the Editor and the Reviewers for their thoughtful comments and constructive suggestions. We have addressed all the points raised by the reviewer and editor comments. The changes in manuscript were highlighted within the document by using the track changes mode in MS Word. Please see the attachment for our response.

Reviewer 2 Report
Vanichanukulyakit and collaborators presented a manuscript on the analysis of mitochondrial DNA in the Thai population to enhance the power of human discrimination by using the mitochondrial DNA.
The manuscript is well written, and contains a lot of data, well shown in the presented tables.
In my opinion the paper is fitted for the journal "Genes".
I only have a minor question for the authors:
The type of polymorphism shown in table 4 in the fourth line (the 34 subjects with the rCRS haplotype) does not seem clearly indicated to me, I suggest the authors check this point.
Author Response

(The authors gave the same response as above.)
